# Regression-Based Normative Data for Independent and Cognitively Active Spanish Older Adults: Free and Cued Selective Reminding Test, Rey–Osterrieth Complex Figure Test and Judgement of Line Orientation

**DOI:** 10.3390/ijerph182412977

**Published:** 2021-12-09

**Authors:** Eva Calderón-Rubio, Javier Oltra-Cucarella, Beatriz Bonete-López, Clara Iñesta, Esther Sitges-Maciá

**Affiliations:** 1SABIEX, Universidad Miguel Hernández de Elche, Avda. de la Universidad, 03207 Elche, Spain; eva.calderon@goumh.umh.es (E.C.-R.); bbonete@umh.es (B.B.-L.); clara.inesta@goumh.umh.es (C.I.); esther.sitges@umh.es (E.S.-M.); 2Department of Health Psychology, Miguel Hernández University of Elche, 03202 Elche, Spain

**Keywords:** mild cognitive impairment, memory, neuropsychological assessment, normative data

## Abstract

The aim of this work was to develop normative data for neuropsychological tests for the assessment of independent and cognitively active Spanish older adults over 55 years of age. Methods: regression-based normative data were calculated from a sample of 103 nondepressed independent community-dwelling adults aged 55 or older (66% women). The raw data for the Free and Cued Selective Reminding Test (FCSRT), the Rey–Osterrieth Complex Figure Test (ROCF) and the Judgement of Line Orientation Test (JLO) were regressed on age, sex and education. The model predicting the FCSRT delayed-recall (FCSRT-Del) scores also included the FCSRT immediate-recall (FCSRT-Imm) scores. The model predicting the ROCF immediate-recall (ROCF-Imm) scores included the ROCF copy-trial (ROCF-C) scores, and the model predicting the ROCF delayed-recall (ROCF-Del) scores included both the ROCF-C and the ROCF-Imm scores. In order to identify low scores, z-scores were used to determine the discrepancy between the observed and the predicted scores. The base rates of the low scores for both the SABIEX normative data and the published normative data obtained from the general population were compared. Results: the effects of the different sociodemographic variables (age, sex and education) varied throughout the neuropsychological measures. Despite finding similar proportions of low scores between the normative data sets, the agreement was irrelevant or only fair-to-good. Conclusions: the normative data obtained from the general population might not be sensitive enough to identify low scores in cognitively active older adults, incorrectly classifying them as cognitively normal compared to the less active population.

## 1. Introduction

The United Nations [1] predicts that, by 2050, the global population of older people will be growing at a rate of 2.6% per year. It is expected that 30% and 6% of the population in developed countries will be aged 60 or older and 80 or older, respectively. In Spain, the number of people aged 55 years or older is around 15 million, which represents 33.08% of the overall population. Amongst these, 18.22% (2,856,102) are 80 years or older [2].

The increase in the population’s age span is related to a raise in the probability of pathological ageing, such as cognitive impairment or dementia [3,4]. Indeed, age is the main risk factor for cognitive impairment [5,6] and Alzheimer’s disease (AD), with the prevalence of AD increasing from age 60 to 89 years [7]. This situation has an important impact on both public health and sanitary and social services [8]. On the one hand, cognitive impairment is a frequent cause of consultation in primary health care, with an estimated prevalence of 15–20% in individuals older than 60 years [9]. In Spain in particular, the prevalence of cognitive impairment is 18.5% in people older than 65 years, and 45.3% in people older than 85 years [8]. On the other hand, the overall prevalence of dementia in the Spanish population is between 4% and 9% in groups older than 65 years [10]. Both cognitive impairment and dementia are related to higher comorbidity [11], susceptibility to infections [12] or depressive symptoms [13]. Furthermore, dementia has the highest total expenditure in neurological diseases [14] and requires about 3 h per day of care in both the basic and instrumental activities of daily life (ADL), becoming a huge burden for the caregivers of a family member with dementia [15].

A high percentage of individuals may not have a formal diagnosis of cognitive impairment or dementia: 55% in Spain [6], 58% in Europe [16] and 65% in the USA [17]. This situation could be due to the patients’ and relatives’ lack of consciousness about the presence of the disease [10], the lack of training among professionals in primary health care or the tight agendas in specialist settings [17]. For this reason, it is necessary to perform a thorough clinical study to overcome these problems. In order to detect subtle deficits or characterise cognitive strengths and weaknesses, an important part of this clinical study would be the neuropsychological assessment [18,19].

A neuropsychological assessment is based on the administration of standardised neurocognitive tests, in order to analyse the cognitive changes after brain damage [18,19]. It is essential to have reliable and suitable normative values, accounting for the effect of sociodemographic and cultural variables, in order to reduce the risk of misdiagnosis [20,21,22], determine the level of performance in the neuropsychological tests [23] and interpret the results obtained by a given subject by contrasting them with the performance of the reference group [18,24]. Normative data can be developed following three different strategies: mean and standard deviation scores, multiple regression and ROC curves. The first strategy is based on determining how the values are distributed and what their associated theoretical or empirical frequencies are [25,26,27]. The second is based on the generation of a prediction model accounting for age, sex and/or education. This model will be used to estimate a score, given certain variables (age, sex and education), that will be compared with the observed data [28,29,30]. ROC curves provide a cut-off value based on sensitivity and specificity [31].

Nevertheless, normative data obtained from the general population might not take into account the characteristics of active ageing. Active ageing is a process in which the opportunities for health, participation and security are optimised in order to enhance the quality of life during ageing [32]. This model encompasses six groups of determinants: behavioural styles, personal biological and psychological conditions, health and social services, physical engagement and social and economic factors. It has been previously reported that engaging in leisure activities is a relevant factor for a well-ageing process [33]. Indeed, most successfully ageing people, characterised by their health and independence, differ from the general population in the number of leisure activities they participate in [34]. Furthermore, cognitively stimulating activities may prevent cognitive decline during ageing [35].

Although educational level may play an important role in active ageing (for instance, [36]), the variable regarded as a determinant factor in the WHO model (WHO, 2002) is not early education, but lifelong learning [37]. In this sense, education is mostly taken into account as a long-term determinant [38]. Hijas-Gómez et al. [37] found that only the physical component was associated with survival and correlated with cognitive status, lifestyle and lifelong learning. For this reason, it is of great importance to ensure equal access opportunities for learning throughout the lifespan [39]. Thus, as active ageing is independent of the educational level, using normative data that account for educational level but do not comprehensively analyse the active ageing model could lead to a higher risk of misdiagnosis.

Two important cognitive domains assessed by neuropsychologists are memory and visuospatial perception [18,24]. In this work, memory was studied with the Free and Cued Selective Reminding Test (FCSRT; [40,41,42]) and the Rey–Osterrieth Complex Figure (ROCF) [43,44]. Visuospatial perception was analysed through the ROCF and the Judgement of Line Orientation (JLO; [45,46,47]) tests.

The FCSRT was initially designed as a selective reminding memory test [40,41], and then a cued-recall trial was added [42]. The ROCF measures the visual perception and visuospatial construction ability by means of the copy trial, and visual memory through the immediate- and delayed-recall tasks [43,44]. For both the FCSRT and ROCF, normative data in a healthy Spanish population have been developed by the NEURONORMA project [48,49]. The JLO analyses spatial perception and spatial orientation [46]. Normative data in a healthy Spanish population for JLO have also been developed by the NEURONORMA project [50].

Since cognitively active people seem to outperform nonactive people in neuropsychological tests, it is likely that the use of normative data that only account for educational level may increase the number of misdiagnoses [51]. For this reason, the objective of this work is to develop normative data for a cognitively active elderly population. We hypothesise that, accounting for the characteristics of the active ageing population, the low scores obtained with the normative data based on cognitively active people (SABIEX) will differ from those of the normative data obtained from the general population (NEURONORMA). Our hypothesis is that the active ageing population will show a higher percentage of low scores in the SABIEX normative data.

## 2. Materials and Methods

### 2.1. Participants

This was a cross-sectional observational study with cognitively healthy individuals living independently in the community. One hundred and five (105) students from the Seniors’ University (*SABIEX*) of the Universidad Miguel Hernández de Elche (UMH, Spain) were recruited from October 2019 to July 2021. SABIEX is an academic university programme for people aged 55 years or older covering subjects such as economics, psychology, politics and the arts.

The features of the sample have been previously described [52]. Inclusion criteria were (a) being 55 years old or older, (b) being cognitively normal (CN) without subjective cognitive complaints and (c) living independently in the community. Cognitive normality was determined by (a) Mini-Mental State Examination (MMSE; [53]) scores higher than 23, (b) Clinical Dementia Rating Scale (CDR; [54]) scores equal to 0 and (c) Lawton–Brody Instrumental Activities of Daily Living (IADL; [55]) scores equal to 7 or higher. Exclusion criteria were (a) unwillingness to participate in the neuropsychological assessment and (b) presence of vision and/or hearing impairments that might have interfered with the administration of cognitive tests. Having past or current medical conditions (e.g., cancer, psychiatric disorders or metabolic disease) was not defined as an exclusion criterion, so that the representativeness of the sample was guaranteed. All participants were born and raised in Spain and had Spanish as their mother tongue.

### 2.2. Materials

The individual neuropsychological assessment spanned around 90 min and covered different cognitive domains. In this paper, data are reported for tests assessing memory and visuospatial perception: the FCSRT [40,41], the ROCF [43,44] and the JLO [45,46,47] tests. Descriptive statistics for the remaining tests can be found in Bonete-López et al. [52].

#### 2.2.1. Free and Cued Selective Reminding Test

The FCSRT is a word-list learning test administered through three immediate free and cued recall tasks and a 30 min delayed free and cued recall task. The FCSRT was administered according to NEURONORMA instructions [49]. Participants were shown 4 different sheets, all of them containing 4 bold written words. Each word pertained to a different semantic category. The four sheets were presented individually and in sequence. Subjects were asked to read each of the four words aloud and match each of them with the semantic clue given by the examiner. After the 16 items were correctly identified, a nonsemantic interference task (counting backwards by three) was carried out for 20 s. After the interference task, the participants were asked to freely recall the 16 words in any order. Ninety seconds were given as the maximum time to complete the task. The semantic category was given (cued recall) for the words not recalled in the free-recall task. This task was followed by two more trials, each of them separated by a nonsemantic interference task (counting backwards by three). In the first two trials, participants were given the correct items for words not recalled with the semantic cue. After 30 min, the delayed-recall trial was performed. If any word was not remembered, the task was followed by the cued-recall trial. Lastly, 40 words were read out loud by the examiner [52]. Of them, 16 were words from the previous free- and cued-recall tasks (targets) and 24 were distractors. Subjects were asked to identify the targets by saying “yes” and the distractors by saying “no”. In this paper, the study variables were the total immediate- and total delayed-recall scores (FCSRT-Imm and FCSRT-Del, respectively) with a maximum score of 48 and 16, respectively.

#### 2.2.2. Rey–Osterrieth Complex Figure

The ROCF test consisted of 18 elements, each of which could be scored 0.5, 1 or 2 points depending on their accuracy and location [24]. The ROCF test included three trials: (a) a copy of the figure model (ROCF-C), (b) an immediate-recall task (ROCF-Imm) and (c) a delayed-recall task after 30 min (ROCF-Del). For each trial, a blank sheet of paper was placed horizontally in front of the participants. In the first trial, participants were asked to copy the figure, trying not to leave any element undrawn and to respect the details and dimensions of the model. They were allowed to rotate their piece of paper, but not the model, which was just shown in the copy trial. For both the immediate- and delayed-recall trials, subjects were asked to draw as many items as they could remember. If they could not remember the location of the element, they were requested to draw it wherever they desired.

The study variables were: (a) copy—overall sum of correctly drawn elements when copying the model; (b) immediate recall—total sum of correctly drawn items after 3 min; and (d) delayed recall—total sum of correctly drawn items after 30 min. The maximum score was 36 for each task.

#### 2.2.3. Judgement of Line Orientation

The JLO test consisted of five practice items and 30 test items. Each element was formed by two unnumbered segments and 11 numbered lines forming a semicircle as a model. The task was based on visually matching both test lines with two of the semicircle’s segments in each trial. The five practice tests were initially presented, and then the task continued with the performance of the test items. The total score was given by the number of elements in which both lines were correctly identified.

### 2.3. Procedure

The participants were invited to voluntarily participate in the study. No credit for their courses were given. The participants were individually assessed by a board-certified neuropsychologist and trained undergraduate, master’s degree-level or PhD-level students. An informed consent form was signed by every subject prior to enrolment. All participants provided personal and familial medical information. This project was approved by the UMH Ethics Committee (DPS.ESM.01.19).

### 2.4. Statistical Analysis

#### 2.4.1. Calculation of Normative Data

We used the methodology reported by Iñesta et al. [56]. Briefly, a linear regression model was built to predict neuropsychological test scores using age, sex and education. To test for possible nonlinear associations, we centered age and education using the lowest value in the distribution of data, which are referred to as Age_Min_ and Education_Min_ throughout the manuscript, and calculated the quadratic term. We used a two-step linear regression model with raw values introduced in the first step and the quadratic terms introduced in the second step.

To improve the interpretation of performance on neuropsychological tasks conditional on other, previous neuropsychological tasks, we calculated normative data for the ROCF-Imm and ROCF-Del tests with a regression model adding ROCF-C scores to demographic variables, which are referred to as ROCF-Imm_SABIEX_ and ROCF-Del_SABIEX_. In the regression model for the FCSRT-Del test, we added FCSRT-Imm scores to the demographic variables. Each variable was centered using the lowest value in the distribution of data.

#### 2.4.2. Comparing Normative Data Sets

The number of low scores shown by our sample when using either the SABIEX or the NEURONORMA normative data was analyzed with the McNemar test (corrected for continuity) for related proportions [57], and the Fleiss’ kappa [57] interrater correlation coefficient for categorical data was used to analyse the level of agreement between normative data sets. According to Fleiss et al. [57], agreement beyond chance can be interpreted as poor, fair to good and excellent for values of 0–0.40, 0.41–0.75 and >0.75, respectively. Low scores were defined as scaled scores equal to or lower than 6 using NEURONORMA, and as z-scores equal to or lower than −1.28 using SABIEX normative data [56]. Using z-scores equal to or lower than −1.28 guarantees that the true positive rate and true negative rate from a linear regression are close to the 95% for a sample of size *n* = 100 or larger [58].

## 3. Results

From a pool of 105 subjects (33% males), two subjects were not included because of MMSE scores <24. Descriptive statistics for the demographic variables and the MMSE, IADL and GSD scores of the 103 participants are reported in Iñesta et al. [56]. A descriptive analysis of the complete test battery can be found in Bonete-López et al. [52].

The differences in age between the sexes were statistically significant (t (df = 101) = 3.06; *p* = 0.004). Specifically, the men were, on average, four years older than the women (men: *M* = 68.47, *SD* = 6.51; women: *M* = 64.42, *SD* = 6.22). No statistically significant differences between the sexes were found in the years of education (t (df = 101) = 0.551; *p* = 0.583), MMSE (t (df = 101) = −1.59; *p* = 0.114), IADL (t (df = 101) = 0.70; *p* = 0.485) and GDS (t (df = 101) = −1.18; *p* = 0.240).

The descriptive statistics for the performance in the different tasks of each neuropsychological test is provided in Table 1.

### 3.1. Calculation of Normative Data

The demographic variables (age, sex and education) showed different effects on each neuropsychological test. However, as no relation was found between the demographic variables and any of the tasks of the ROCF test, normative data were calculated with the means and standard deviations.

The multiple linear regression models of the rest of the neuropsychological measures are shown in Table 2. Sex was significantly related to the FCSRT-Imm, FCSRT-Del and JLO scores. Education_Min_ was associated with the FCSRT-Imm, FCSRT-Del and FCSRT-Del_SABIEX_ scores. Education_Min_^2^ had an effect on the FCSRT-Del and JLO scores. No statistically significant effects of Age_Min_ and Age_Min_^2^ were found. Particularly interesting for this work are the significant associations within the dependent tasks, as the FCSRT-Del scores were related to the FCSRT-Imm_Min_ scores, the ROCF-Imm scores to the ROCF-C_Min_ scores and the ROCF-Del scores to the ROCF-Imm_Min_ scores.

### 3.2. Comparing Normative Data Sets

The frequency and cumulative percentage of the low scores for the three normative data sets (NEURONORMA, SABIEX_INDEP_ and SABIEX_DEP_) are provided in Table 3 and Figure 1.

No statistically significant differences were found between the NEURONORMA and SABIEX_INDEP_ data sets in the percentage of participants with one or more low scores (*χ*^2^ (*N* = 103) = 0.842; *p* = 0.358). The Fleiss’ kappa coefficient showed only a fair-to-good agreement for both data sets when identifying the participants showing one or more low scores (κ = 0.586; *p* < 0.001).

Conversely, statistically significant differences were found in the proportion of participants showing one or more low scores between the NEURONORMA and the SABIEX_DEP_ data sets (*χ*^2^ (*N* = 103) = 6.50; *p* = 0.011). Furthermore, although the Fleiss’ kappa coefficient was statistically significant, the agreement between the normative data sets in identifying the low scores was only fair-to-good (κ = 0.464; *p* < 0.001).

### 3.3. Free and Cued Selective Reminding Test

Using the SABIEX normative data, 8.74% of the sample had at least one low score, whereas for the NEURONORMA data, 1.94% of the sample (two subjects) obtained one or more low scores. The McNemar test was statistically significant (*χ*^2^ (*N* = 103) = 5.143; *p* = 0.023), meaning that the proportions of low scores were not similar in both data sets. The Fleiss’ kappa coefficient showed a lack of agreement in identifying the low scores (κ = 0.328; *p* = 0.001) (see Appendix A).

We also performed separate analyses of the proportion of low scores for the neuropsychological measures considered dependent from the others and compared them to the NEURONORMA data set. First, we compared both the FCSRT-Del and FCSRT-Del_SABIEX_ scores with the data from NEURONORMA. In both the FCSRT-Del and FCSRT-Del_SABIEX_ data, 11.65% of the sample (12 subjects) had at least one low score, in contrast to the 0.97% (one subject) from NEURONORMA. The McNemar test showed statistically significant differences in the proportion of both the SABIEX and NEURONORMA data sets (*χ*^2^ (*N* = 103) = 9.091; *p* = 0.002) and no agreement was found in identifying the low scores (κ = 0.097; *p* = 0.326) (see Appendix A).

### 3.4. Rey–Osterrieth Complex Figure

The ROCF-C scores were not associated statistically with any of the variables analysed (see Table 2). Regarding the ROCF-Imm and ROCF-Imm_SABIEX_ scores, at least one low score was obtained by 10.68% of the sample (11 subjects) in the SABIEX_INDEP_, 8.74% (9 subjects) in the SABIEX_DEP_ and 6.80% (7 subjects) in the NEURONORMA data set. The McNemar test showed no statistically significant differences when comparing the NEURONORMA data with either the SABIEX_INDEP_ (*χ*^2^ (*N* = 103) = 1.500; *p* = 0.221) or the SABIEX_DEP_ (*χ*^2^ (*N* = 103) = 0.125; *p* = 0.7237) data. The Fleiss’ kappa coefficient showed fair-to-good agreement between the NEURONORMA data and both of the SABIEX data sets (SABIEX_INDEP_: κ = 0.635; *p* = 0.000; SABIEX_DEP_: κ = 0.458; *p* < 0.001) (see Appendix A).

Regarding the ROCF-Del and ROCF-Del_SABIEX_ scores, at least one low score was obtained by 12.62% of the sample (13 subjects) in the SABIEX_INDEP_, 10.68% (11 subjects) in the SABIEX_DEP_ and 9.71% (10 subjects) in the NEURONORMA data set. The McNemar test showed no statistically significant differences when comparing the NEURONORMA data with either the SABIEX_INDEP_ (*χ*^2^ (*N* = 103) = 0.800; *p* = 0.371) or the SABIEX_DEP_ (*χ*^2^ (*N* = 103) = 0.000; *p* > 0.999) data sets. However, the Fleiss’ kappa coefficient was only statistically significant when comparing the NEURONORMA data with the SABIEX_INDEP_ data (κ = 0.755; *p* = 0.000), with a lack of agreement in identifying the participants as showing one or more low scores between the SABIEX_DEP_ and NEURONORMA data sets (κ = −0.114; *p* = 0.249) (see Appendix A).

### 3.5. Judgement of Line Orientation

The JLO scores were negatively associated with the sex variable and positively associated with the Education_MIN_^2^ variable, meaning that the effect of education was not linear and that better scores are expected when the level of education increases.

## 4. Discussion

The objective of this study was to develop normative data for Spanish cognitively active individuals aged 55 or older. For this aim, regression analyses were performed for every variable studied. Two novelties characterise this work: the use of a cognitively active sample and the analysis of every subtest as both dependent (SABIEX_DEP_) and independent (SABIEX_INDEP_) of other variables from the same test. Our results showed only a fair-to-good agreement in identifying the low scores between both the SABIEX normative data and the NEURONORMA normative data, with no agreement for the delayed-recall trial from the ROCF test.

No effect of age was found in any of the tests assessed, which differs from previous studies suggesting that neuropsychological performance declines with advancing age [49,59,60,61,62]. This result may represent a novelty in this area, as it may show that age becomes less relevant when individuals are cognitively active. This may also evidence the importance of creating a cognitive reserve [63] throughout the lifespan, independently of the level of education achieved during the early life stages.

Regarding the FCSRT, sex had a significant effect on both the FCSRT-Imm and FCSRT-Del trials, in line with previous studies suggesting that women outperform men on tests based on verbal material such as the FCSRT [64,65,66]. Other studies also showed a significant effect of sex in the free-recall tasks of the FCSRT [67,68]. However, its effect may be minor or irrelevant [49,69,70] and may be influenced by the different proportion of men–women in our study. The effect of sex became non-significant when we analysed the FCSRT-Del scores while controlling for the FCSRT-Imm scores (FCSRT-Del_SABIEX_). Education also showed a significant effect on the FCSRT-Imm, FCSRT-Del and FCSRT-Del_SABIEX_ scores, in line with previous studies [49,66,67,68,69,70]. This result may illustrate the effect of education on neuropsychological performances and its association with cognitive reserve [61,69,70,71]. Quadratic education had an impact only on the FCSRT-Del scores, meaning that the effect of education was not linear and that the differences in memory performance were more significant amongst less educated subjects than between highly educated individuals, which is in line with previous works [72,73,74,75]. This effect disappeared when the FCSRT-Imm score was also controlled.

Regarding the ROCF test, no significant effect of any of the sociodemographic variables was found. In previous studies, minor or nonexistent effects have been reported [49,76,77,78], which is in line with our results. However, most of the scientific research shows that education has a positive impact when performing ROCF tasks, as higher education levels are usually associated with better scores [49,78,79,80]. Nevertheless, our results may indicate that, independently of the level of education, when a cognitively active lifestyle is carried out, cognitive strategies can compensate for these differences [81], as suggested by the cognitive reserve construct [63,82]. Lastly, regarding the JLO test, sex and quadratic education showed a significant effect, which is in line with previous studies [83,84,85].

Another relevant aspect of our research was the analysis of the different subtests as both independent of and dependent on the rest of the trials of the same test. In line with previous results [78], the ROCF-C scores had a positive effect on the ROCF-Imm scores and the ROCF-Imm on the ROCF-Del, and the FCSRT-Imm scores had a significant positive effect on the FCSRT-Del scores. When comparing the SABIEX_INDEP_ and NEURONORMA data sets, we found similar proportions of low scores, but there was only a fair-to-good agreement in identifying the low scores. On the other hand, when comparing the SABIEX_DEP_ and NEURONORMA data sets, we found a statistically different proportion of low scores and only a fair-to-good agreement, which was even lower than that of the SABIEX_INDEP_ data set. This may have been due to the distinctive features characterising our sample: the NEURONORMA data was based on the general population, while the SABIEX sample included highly cognitively active participants. This may suggest that using normative data that do not take into account the specificities of the active ageing population may increase the number of diagnosis errors and, hence, the misclassification of subjects as cognitively impaired if low scores are to be used to diagnose cognitive impairment. Additionally, not controlling the effect of the related subtests when creating new normative data sets, as is the usual practice [49,79,80,84], may also be associated with an increased rate of false positives and/or negatives and with the misclassification of subjects. Further research is needed to confirm the clinical applicability of our results, analysing whether the use of SABIEX normative data is useful to identify with greater certainty individuals with a greater risk of cognitive decline. Lastly, when contrasting the neuropsychological measures separately, no agreement (FCSRT-Imm, FCSRT-Del, FCSRT-Del_SABIEX_, ROCF-Del_SABIEX_) or only a fair-to-good agreement (ROCF-Imm, ROCF-Imm_SABIEX_, ROCF-Del) was found in identifying low scores. This supports the idea that normative data accounting for the characteristics of cognitively active elders are needed to avoid the appearance of false negatives and/or false positives and, consequently, their misclassification.

These results may have important clinical implications. For example, objective cognitive impairment (i.e., one or more low scores) is necessary for the diagnosis of MCI [86,87]. Our results are important whether classic [86,88] or modified [89,90,91] diagnostic criteria are to be used to identify MCI. If only one test is used to identify memory impairments, as with the classic criteria, the selection of the appropriate normative sample against which the raw scores are to be compared may reduce the number of false positives. If several tests are used to identify MCI, as with the modified diagnostic criteria, the association between several tasks within the same test might identify with greater certainty individuals with true cognitive impairment. For example, if two low scores on the memory task are needed to meet the criterion for the comprehensive diagnostic criteria [89,92], using the delayed-recall scores as independent of the immediate-recall scores might be associated with a higher number of false positives, because the correlation between the tests is not taken into account when independent normative data are used. Relatedly, analysing the performance on some tests that are conditional on other tests might impact the number of low scores needed to define normal variability and, subsequently, true cognitive impairment [90,91]. Using the worst-performing 10% of the sample, two or more low scores would be needed to meet the criterion for objective cognitive impairment using the NEURONORMA normative data, compared to three or more low scores using the SABIEX normative data (Table 3). Additionally, since low scores on visual memory tasks have been associated with a similar risk of progression from MCI to AD compared to low scores on verbal memory tasks [93], using the normative data for the delayed-recall trial in the ROCF test conditional on the copy trial may help clinicians to better differentiate true impairments in visual memory from low scores on the immediate or delayed trials that are conditional on low scores on the copy trial. If low scores on the delayed-recall task are associated with low scores on the copy trial, the individual would be diagnosed as having nonamnestic MCI rather than as having amnestic MCI, with the former being associated with a lower risk of progression from MCI to AD than the latter [94]. We provide a friendly calculator for researchers and clinicians that is available at https://docs.google.com/spreadsheets/d/1p-RDT6F85EsXPxALV-l6R8Sq-E4qGEr0/edit?usp=sharing&ouid=109893127231470805500&rtpof=true&sd=true (accessed on 8 December 2021). 

This work has certain limitations. First, we used an incidental sampling method, which was related to the overrepresentation of people aged 60–70 and the underrepresentation of people aged over 75 years. It was also associated with the overrepresentation of women. As gender differences in cognitive domains have been previously suggested [95,96], these results should be interpreted with caution. The number of younger and older subjects, as well as the number of men, may be further investigated to improve the reliability and accuracy of the data set. Secondly, with our methodology, we cannot determine the clinical applicability of the SABIEX normative data. We can only suggest that individuals showing low scores might differ between the SABIEX and population-based normative data sets, but further research into the clinical population is needed to find out which is more accurate in identifying cognitive impairment in cognitively active older adults if low scores are to be used to identify cognitive impairment. Lastly, this work was only based on the Spanish population, so, due to the cultural differences in neuropsychological functions suggested by other authors [21,22,97], interpretations and application in other cultures or languages should be taken with caution.

## 5. Conclusions

Our results suggest that normative data obtained from the general population might not be sensitive enough to identify low scores in cognitively active older adults. Normative data accounting for the characteristics of cognitively active older adults might be necessary to reduce the number of diagnostic errors and, eventually, their misclassification as cognitively impaired compared to normative data obtained from the general and less cognitively active population.

## Figures and Tables

**Figure 1 ijerph-18-12977-f001:**
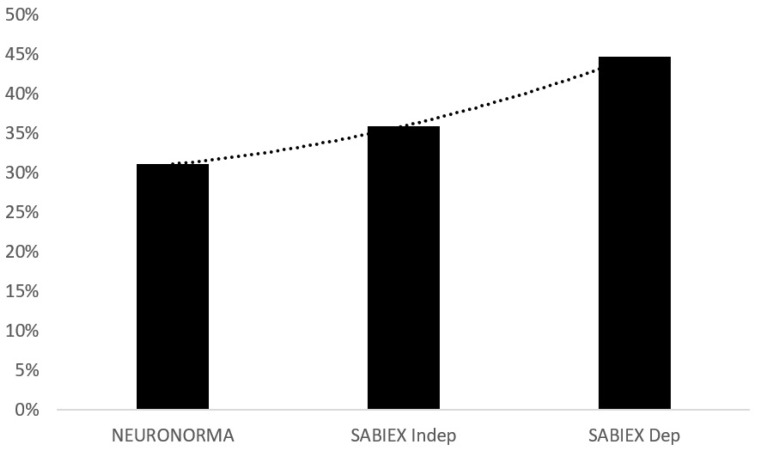
Percentage of participants with one or more low scores per normative data set.

**Table 1 ijerph-18-12977-t001:** Descriptive statistics for neuropsychological performance.

Neuropsychological Test	M	SD	Range
FCSRT-Imm	44.01	4.04	26–48
FCSRT-Del	14.86	1.48	9–16
ROCF-C	28.18	2.24	14–36
ROCF-Imm	14.97	4.90	2–26.5
ROCF-Del	15.04	4.80	2–25
JLO	20.55	4.94	9–30

M: mean; SD: standard deviation; FCSRT-Imm: Free and Cued Selective Reminding Test, total immediate recall; FCSRT-Del: Free and Cued Selective Reminding Test, total delayed recall; ROCF-C: Rey–Osterrieth Complex Figure, copy; ROCF-Imm: Rey–Osterrieth Complex Figure, immediate recall; ROCF-Del: Rey–Osterrieth Complex Figure, delayed recall; JLO: Judgement of Lines Orientation.

**Table 2 ijerph-18-12977-t002:** Multiple linear regression models.

		B	Std. Error	Sig.	R^2^	SEE
FCSRT-Imm	Intercept	40.31	1.17	0.000	0.104	3.865
	Education_Min_	0.288	0.111	0.011		
	Sex	1.886	0.811	0.022		
FSCRT-Del	Intercept	11.567	0.836	0.000	0.195	1.350
	Education_Min_	0.547	0.185	0.004		
	Sex	0.667	0.283	0.021		
	Education_Min_^2^	−0.021	0.010	0.030		
FSCRT-Del_SABIEX_	Intercept	9.386	0.430	0.000	0.632	0.908
	FCSRT-Imm_Min_	0.272	0.023	0.000		
	Education_Min_	0.069	0.027	0.011		
ROCF-Imm_SABIEX_	Intercept	8.771	1.57	0.000	0.144	4.554
	ROCF-C_Min_	0.872	0.044	0.000		
ROCF-Del_SABIEX_	Intercept	3.727	0.616	0.000	0.792	2.199
	ROCF-Imm_Min_	0.872	0.044	0.000		
JLO	Intercept	22.252	0.958	0.000	0.208	4.446
	Sex	−4.263	0.933	0.000		
	Education_Min_^2^	0.014	0.007	0.038		

FCSRT-Imm: Free and Cued Selective Reminding Test, total immediate recall; FCSRT-Del: Free and Cued Selective Reminding Test, total delayed recall, independently calculated from FCSRT-Imm; FCSRT_Del_SABIEX_: Free and Cued Selective Reminding Test, total delayed recall, conditional on FCSRT-Imm; ROCF-Imm_SABIEX_: Rey–Osterrieth Complex Figure, immediate recall, conditional on ROCF-C; ROCF-Del_SABIEX_: Rey–Osterrieth Complex Figure, delayed recall, conditional on ROCF-C and ROCF-Imm; JLO: Judgement of Line Orientation.

**Table 3 ijerph-18-12977-t003:** Frequency and accumulated percentage of NEURONORMA, SABIEX_INDEP_ and SABIEX_DEP_ low scores.

Low Scores	NEURONORMA		SABIEX_INDEP_		SABIEX_DEP_	
	Freq	Cum%	Freq	Cum%	Freq.	Cum%
0	71	100	66	100	57	100
1	23	31.1	17	35.9	31	44.7
2	3	8.8	12	19.5	12	14.6
3	4	5.9	5	7.8	2	2.9
4	1	2	2	2.9	1	1
5	1	1	1	1	0	0
Total	103		103		103	

Freq.: frequency of subjects showing the number of low scores in the *Low Scores* column. Cum%: cumulative percentage.

## Data Availability

Data are available upon request from the corresponding author.

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
