# Peer review of "Regression-Based Normative Data for Independent and Cognitively Active Spanish Older Adults: Free and Cued Selective Reminding Test, Rey–Osterrieth Complex Figure Test and Judgement of Line Orientation"

_ijerph, 2021, doi:10.3390/ijerph182412977_

Round 1

Reviewer 1 Report

How did you arrived at the sample size?

Author Response

We added a text with a reference explaining that a sample of n=100 or larger is sufficient to get a true positive rate and a true negative rate close to 95% using a linear regression and residual z-scores ≤-1.28.

Reviewer 2 Report

Congratulation for the wonderful paper. It is recommended that type of study appears in the title. The data in the tables appears duplicated in the text, please delete it. The conclusion must be clear and concise, and it must answer the main objetive of the research. It should not extend more than 2-3 lines.

Author Response

We removed some repeated text in the manuscript and modified the conclusion section accordingly. We used the same title as in the paper by Iñesta et al. (2021). As both papers are part of the same project, we would be grateful if we could retain the title as is.

Reviewer 3 Report

The manuscript is well written and explores an important question that could have critical implications for future investigations in many fields. Another strength is that the authors included a large amount of relevant references giving the readers sufficient context for the study and its observed findings. Having said this, I have some comments I hope the authors could address.

  1. I understand that the method was previously discussed elsewhere (in a published paper), however, I believe that more information should also be included in the current manuscript for the readers to be able to understand and replicate the study.
  2. Could the authors include a couple of figures showing the observed findings? I strongly believe that the manuscript will greatly benefit from including some result figures.
  3. In the discussion, can the authors include some statements on how they think the current findings lay foundation for future studies that wish to investigate relevant topics? What are some future directions that can naturally follow the reported findings?

Author Response

  1. We believe that all relevant information is included in this manuscript. The method used to build the regression equations and how to interpret the participants’ performance is explained in detail in this manuscript. The readers are referred to other papers for information on some other tests in the neuropsychological battery (e.g., MMSE) that are not part of the work reported in this manuscript.
  2. We included a figure showing the difference in the percentage of participants showing one or more low scores among normative data sets, in line with our primary hypothesis.
  3. We included a paragraph discussing possible implications of our results for the diagnosis of MCI and the identification of the risk of progression from MCI to AD.

Round 2

Reviewer 3 Report

I am satisfied with the authors' responses to my previous comments.